# Network Analysis for a Community-Based School- and Family-Based Obesity Prevention Program

**DOI:** 10.3390/healthcare10081501

**Published:** 2022-08-09

**Authors:** Katharina Brauer, Hagen Wulff, Sabine Pawellek, Alexandra Ziegeldorf

**Affiliations:** 1Institute of Exercise and Public Health, University of Leipzig, Jahnallee 59, 04109 Leipzig, Germany; 2Department of Sport and Health Sciences, Faculty of Human Sciences, University of Potsdam, Karl-Liebknecht-Str. 24–25, 14476 Potsdam, Germany

**Keywords:** public health, children, stakeholder, collaboration, network analysis

## Abstract

Rising childhood obesity with its detrimental health consequences poses a challenge to the health care system. Community-based, multi-setting interventions with the participatory involvement of relevant stakeholders are emerging as promising. To gain insights into the structural and processual characteristics of stakeholder networks, conducting a network analysis (NA) is advisable. Within the program “Family+—Healthy Living Together in Families and Schools”, a network analysis was conducted in two rural model regions and one urban model region. Relevant stakeholders were identified in 2020–2021 through expert interviews and interviewed by telephone to elicit key variables such as frequency of contact and intensity of collaboration. Throughout the NA, characteristics such as density, centrality, and connectedness were analyzed and are presented graphically. Due to the differences in the number of inhabitants and the rural or urban structure of the model regions, the three networks (network#1, network#2, and network#3) included 20, 14, and 12 stakeholders, respectively. All networks had similar densities (network#1, 48%; network#2, 52%; network#3, 42%), whereas the degree centrality of network#1 (0.57) and network#3 (0.58) was one-third higher compared with network#2 (0.39). All three networks differed in the distribution of stakeholders in terms of field of expertise and structural orientation. On average, stakeholders exchanged information quarterly and were connected on an informal level. Based on the results of the NA, it appears to be useful to initialize a community health facilitator to involve relevant stakeholders from the education, sports, and health systems in projects and to strive for the goal of sustainable health promotion, regardless of the rural or urban structure of the region. Participatory involvement of relevant stakeholders can have a positive influence on the effective dissemination of information and networking with other stakeholders.

## 1. Introduction

‘Complex problems’ describe a mechanism in which multiple factors and actors are involved and interconnected either through direct or indirect links. This multiplicity of connections makes it impossible to predict the effect of a single intervention [1]. Therefore, solving complex problems requires multiple intervention approaches that go beyond the focus of personal responsibility [2]. Applied to the context of public health, two of the major problems of the 21st century affecting children worldwide can be identified: overweight and obesity. In general, overweight and especially obesity in children and adolescents are associated with adverse health outcomes in the long term [3] with a higher risk of developing non-communicable diseases or co-morbidities and premature death in adulthood [4]. Rising childhood obesity with its detrimental health consequences is a multifactorial phenomenon and poses a challenge to the health care system [5] as its complexity leads to a continuous expansion of this problem [6].

To date, diverse approaches to obesity prevention have been applied that focus on single factors, such as the canteen menu in the school setting or parenting skills in the family unit [7]. However, this approach has largely proven ineffective [8], resulting in a shift from individual-focused interventions to system-wide, community-based interventions, which are emerging as promising [9,10]. To understand the complexity of health behaviors such as physical activity or obesity, systems approaches are recommended to depict the interactions between factors, including the bidirectionality of their influence [11]. Community-based interventions should be designed with a focus on the target group and appropriate community penetration in order to achieve comprehensive behavior change, which begins at the individual level and depends on reinforcement and acceptance at the community level [12].

Speaking of different levels and health, an approach from social ecology immediately comes to mind, namely the rainbow model of Dahlgren and Whitehead [13], which identifies different determinants of individual health that operate in the context of different settings and levels. Networks run throughout this model, but the question arises as to where they are found, what they look like, and who exactly is involved. There is often an ideal–typical categorization of network levels into primary (micro level, i.e., individual), secondary (meso level, i.e., local associations, public institutions, and organizations), and tertiary (macro level, i.e., institutions at the superordinate or local government level) networks that determine health [14] and originate from the ecosystem theory of Bronfenbrenner [15]. In order to promote and strengthen health in the sense of the socio-ecological approach, network characteristics and the interaction of the three network levels with its actors must be identified [14]. This can be accomplished by including different settings (multi-setting approach) as well as different key actors (participatory approach), which, as an example, was performed by Koorts, et al. [16] in their umbrella review applying several systems approaches to illustrate how determinants influencing active leisure time and responsible actors are linked across multiple levels of influence in the ecosystem. Key actor involvement is very important as key actors can play a positive and proactive role in changing factors that put the target population at risk of unhealthy behaviors [12].

To engage stakeholders, these key actors must first be identified by means of a stakeholder analysis. Subsequently, the stakeholder network’s structure can be explored using network analysis. Network analysis serves as a suitable tool to gain insights into the complex structural and processual characteristics of stakeholder networks within community-based interventions [17] and can assist in developing an understanding of the roles that stakeholders play in the outcomes of a network. Suitable measures for assessing network characteristics are density and centrality, while the interaction of actors is surveyed in terms of frequency and intensity of contact [18].

To date, several studies have analyzed networks in the context of community obesity prevention interventions [2,19,20,21,22,23]. However, the networks often consisted only of stakeholders from higher network levels, such as steering committees, local governments, or institutions delivering obesity prevention initiatives. Since it is assumed that successful health promotion occurs through the interaction of the aforementioned health-related network levels, it is important to capture the type and extent of stakeholder interaction. The objective of this research was to identify the network of relevant stakeholders from all network levels for the project ‘Familie+—Gesundes Zusammenleben in Familie und Schule’ (‘Family+—Living healthily together in family and school‘). ‘Familie+’ is based on a community-based, participatory approach aimed at preventing obesity in primary school children in German municipalities. The multi-setting approach involves the joint implementation of evidence-based measures to change weight-related behavior (physical activity, sedentary behavior, nutrition, sleep) in the settings important to children, including family and school. The respective measures will be selected, adapted, and implemented in the two settings (family and school) in a participatory approach together with the relevant stakeholders. This paper applies network analysis to analyze network relationships among stakeholders potentially relevant to the project as well as participating elementary schools in three municipalities in Germany. In particular, structural and processual characteristics were used to identify community network structures and to map existing relationships and collaborations. The three model regions were compared against the background of different community structures (urban versus rural) and varying levels of prior experience in the field of community health promotion. The aim of the study was to use network analysis as an applicable tool for evaluating stakeholder networks at the baseline of a community-based prevention intervention. With the results obtained, the network structures and possible differences in the participating model regions can be mapped, which may be profitable in terms of derivations for the further course of the intervention.

## 2. Materials and Methods

### 2.1. Subjects and Setting

This study included a total of three model regions in Germany as part of the ‘Familie+’ project. Two of them were rural with populations of about 143,000 (network#1) and 211,000 (network#2), and the third region was urban with a population of 590,000 (network#3). Community health promotion is established differently in all three model regions. It is most pronounced in the urban region (network#3), whereas it has been moderately established within one rural region (network#1) and is still in its early stages in the other rural region (network#2).

### 2.2. Network Analysis: Preparation

To identify potentially relevant stakeholders for the research project ‘Familie+’ in the selected municipalities, a stakeholder analysis was carried out in the period May 2020–July 2020. Following the preliminary theoretical work and a literature search for similar research projects in scientific databases, expert interviews following Mieg and Näf [24] were conducted. Meta-level experts were defined as individuals who have at least ten years of professional experience in the setting of community health promotion and/or have initiated and implemented at least two community health promotion projects. Experts in the community setting were defined as persons who have several years (at least five) of professional experience in the community health promotion setting and/or have initiated and/or implemented and/or accompanied at least one project on community health promotion. Based on this, individual expert interviews were conducted to identify all potential stakeholders in the three selected municipalities. All interviews were guided and conducted by telephone (due to the COVID-19 pandemic) and fully transcribed and analyzed according to an extended simple transcription system by Dresing and Pehl [25]. Thereafter, all interviews were analyzed using qualitative content analysis according to Mayring [26], and the results on all potential stakeholders are presented on the meta-level as well as separately for the three model regions in individual mind maps.

After identifying all potentially relevant stakeholders, the network analysis was conducted separately for the three selected model regions in the time period November 2020–January 2021. Subsequently, data entry and evaluation took place from February 2021 to June 2021.

### 2.3. Network Analysis: Execution

To establish initial contact with all stakeholders identified as potentially relevant in the stakeholder analysis, a network analysis was conducted. To capture the type and extent of stakeholder interactions from all network levels, the strategy of examining networks as a whole was chosen, analyzing what types of relationships each actor in a given study set of actors (here: model region) maintains or does not maintain with every other actor in that set [27]. In preparation for the potentially relevant stakeholders, e-mails were sent with information about the project, the privacy policy, and the network analysis procedure. Three to five days later, each stakeholder received a call from an employee of the University of Leipzig to arrange an appointment to conduct the telephone survey. For all telephone surveys, institution-specific questionnaires were developed to keep track as a surveyor as well as to ensure that institutions could not rate the relationship to themselves. For example, if 30 institutions were identified within a community, all 30 institutions were asked to answer the questions posed with respect to the other 29 institutions. During the interview, the questions were systematically worked through and noted on the questionnaire by the trained interview guide. Afterwards, the data were anonymized by coding and stored in accordance with data protection regulations. Following Schoen, et al. [28] and Schnegg and Lang [27], the survey included a series of questions to assess several key variables, including frequency of contact, level of collaboration [29], communication channels, and objects of communication, in the context of municipal health promotion with the respective network partners. Participants were presented with a roster of names of other institutions in their network and reported on key variables for each institution. They were first asked whether they knew the institution and if they generally had contact with it. These questions served as a basis for the later analysis of network density and centrality measures from which statements about the individual integration of an actor into the overall structure can be derived [27]. If the answer was ‘yes’, they were asked on a six-point scale how often they had direct contact with each of the other people in their network about community health promotion in the past year (no contact, yearly, quarterly, monthly, weekly, and daily). Collaboration was assessed with a scale adapted from established network analytic methods [30]. Participants were asked to select the response that best describes the relationship in the last year with each of the institutions on the roster. Response options ranged from not linked (do not work together) to communication (share information only), cooperation (work together informally to achieve common goals), collaboration (work together as a formal team with specific responsibilities), and fully linked (work together as a formal team and mutually plan and share staff or resources to accomplish goals).

### 2.4. Network Analysis: Evaluation

The networks are presented separately by model region. To examine structural patterns, network analysis represents key actors [27] as the smallest unit (node) in the network [31], while social relationships are visualized as ties between nodes, constituting the main focus of a network [32]. The relationships of potentially relevant stakeholders within a network were mapped in the form of a matrix that gives information about the relationship between two actors. If the upper and lower halves of the diagonal are identical, the matrix is called symmetric and the relation of one actor is reciprocated by its counterpart. Furthermore, a distinction is made between weighted and unweighted matrices; in the case of an unweighted matrix, statements are made about the (non-)presence of a relationship, but no statements are made about the intensity of the relationships [27]. Furthermore, network analysis can be used to represent process characteristics, predict inter-organizational communication and collaboration, and measure partnership characteristics in community health promotion [18,33]. Process characteristics such as density (the number of actually existing connections expressed as a proportion of all possible relationships), centrality (degree centrality: the number of incoming and outgoing relationships of an actor; betweenness centrality: an indicator of the influence of an actor in a network that gives information about the extent of its control potential), homophily (similarity of actors with respect to certain characteristics or attributes; values between −0.5 and −1 = homophilic, values between 0.5 and 1 = heterophilic, values between −0.5 and 0.5 = neutral), and connectedness (frequency and intensity of contact) were examined [27,34].

All analyses were performed with UCINET 6.0 software and the NetDraw network visualization tool was used to generate graphical representations of each network [35].

## 3. Results

A total of 69 potentially relevant stakeholders from the three model regions were contacted to participate in the network analysis. Overall, the response rate was 67%, with individual response rates of 65% for network#1, 70% for network#2, and 67% for network#3. The characteristics of each model region in terms of the number of potential stakeholders, associated structural level, and subject area are shown in Table 1. This table provides a very comprehensive overview and lists all stakeholders as sequential numbers for each network, which will be relevant for the following figures.

The network analysis was conducted separately by model region and the results are presented in more detail in the following sections.

### 3.1. Network Characteristics

Table 2 shows the network structure summaries. All measures are for undirected ties, which means that statements are only made about the (non-)existence of a relationship, but not about the intensity of the relationships. On average, network#1 was bigger and had a higher average degree than network#2 and network#3. The relationships of two actors are reported from both positions per network, meaning that, in total, there are 400 possible contacts in network#1 with 20 actors, for example.

Network#1 has the highest average degree with an average number of nine links. Network#2 and network#3 follow, matching the decreasing network size, with average links per node of nearly seven and nearly five, respectively.

The network density of network#2 is 0.52, meaning that this network contains more than half (52%) of the possible edges expected in a completely interconnected network. Compared with the other networks with 48% for network#1 and 42% for network#3, network#2 has the highest density with its network size of 14 institutions.

The average path length within network#1 and network#2 is very similar. The length of network#3 is slightly longer at 1.64, meaning that, on average, institutions can reach each other by following 1.64 causal paths.

Degree distributions of all networks are shown in Figure 1.

Since there are three non-symmetric networks, the in-degree and out-degree values of the individual institutions are not uniform. The distribution of the node in-degree ranges from 4 to 16 for network#1, from 1 to 12 for network#2 and from 2 to 9 for network#3. For network#1 and network#2, the institution with the highest in-degree value is a social and educational institution from the local level with a focus on health, whereas for network#3 the most popular institution with the highest value is a higher, communally organized institution with a focus on health. This institution also has the highest out-degree value, whereas in network#1 one institution at the network level in the field of nutrition and in network#2 one of the elementary schools along with a social and educational institution with a focus on health at the local level have the highest level of expansivity within the network.

The degree centrality is very similar for network#1 and network#3 at 0.57 and 0.58, respectively, and is one-third larger compared with network#2 with a degree centrality of 0.39.

All three networks are heterophilic with respect to the feature subject area (network#1, 0.61; network#2, 0.69; network#3, 0.57). Regarding the structural level, no homophily or heterophily can be identified; therefore, the networks are neutral.

A visual representation of the diversity of the analyzed networks can be found in Figure 2. For an overview of all stakeholders and their subject area, see Table 1.

All relationships were considered to be unidirectional and therefore are represented as directed graphs. The node size depends on the betweenness centrality, meaning that the larger the nodes, the shorter their geodesic paths of connections to all other nodes and the more central the respective actor. In network#2 and network#3, one institution each from the subject area ‘health’ has the greatest betweenness centrality, whereas the most central actors within network#1 are a social and educational institution together with an actor in the field of nutrition.

### 3.2. Frequency of Contact

As the largest network studied, network#1 had the highest average degree (9.2) with its 151 out of 400 possible contact opportunities. Putting non-contacts in the background, the most frequent form of contact in all three networks is quarterly, with quarterly contact in network#3 accounting for two-thirds of the total contact within the network. Remarkable in network#2 is the strong annual and daily contact between the institutions compared with the other two networks (see Figure 3).

### 3.3. Intensity of Collaboration

The most frequent intensity of collaboration in all three networks was informal collaboration to achieve common goals, which in the case of network#3 even accounted for more than half of the collaboration types. Considering the four types of existing collaboration, network#1 stands out with more than one fifth of the strongest collaboration as a formal team (joint planning/sharing of staff and resources), although the network size should not be underestimated (see Figure 4).

### 3.4. Visualization of all Networks

A visual representation of the analyzed networks regarding the frequency of contact (see Figure 3) can be found in Figure 5. For an overview of all stakeholders and their subject area, see Table 1. 

A visual representation of the analyzed networks regarding the intensity of collaboration (see Figure 4) can be found in Figure 6. For an overview of all stakeholders and their subject area, see Table 1.

## 4. Discussion

In this study, network analytical methods were used to give an overview of the three model regions involved in the project ‘Familie+’. Their structural and process characteristics as well as the interconnectedness of the actors within each network were outlined. The networks were mainly characterized by their different sizes as well as different distributions of network partners in terms of their subject matter and structure. Network partners who are in contact interact with each other mostly on a quarterly basis and in the context of informal collaboration. Social networks are particularly important in terms of supporting partnerships and collaborations within a research process [36].

If the three networks are placed in the context of their structure for community health promotion, the following characteristics can be identified. Although network#2 for community health promotion is still in its early stages, it is intriguing to note that it has both the highest network density and the lowest degree centrality. In contrast, network#3, with its already very developed structure for community health promotion, has the lowest network density and, similar to network#2, one-third greater degree centrality than network#3. High degree centrality can have both benefits and drawbacks, as a small number of stakeholders connect many other institutions that would otherwise not be connected. This can have a positive impact on the effective dissemination of information and strengthen networking with other institutions through these key stakeholders, but, at the same time, communication depends on these gatekeepers, who could represent bottlenecks regarding the transfer of information [37]. In network#2 and network#3, one institution from the subject area ‘health’ has the greatest betweenness centrality and represents an important communication interface. Betweenness centrality is an indicator of an actor’s influence in a network and provides information about the extent of its control potential. Actors with high betweenness centrality serve as a strong link between other network partners through their strategic network placement and serve, for example, as mediators of information [27,32]. In network#1, on the other hand, the most central actors are a social and educational institution and an actor in the field of nutrition. In this region, however, there is a greater number of stakeholders dealing with the subject area ‘health’ on a superordinate level compared with the other two regions, which in turn could explain the lower centrality of these institutions. When analyzing networks as a whole, adjusting for network size is important because the larger the network, the less dense it becomes and the higher the degree of centralization [18]. Especially with regard to the high degree centrality in #network1 and #network3, the very short average path length in all three networks represents an optimal condition for the implementation of community health promotion projects, as most relevant institutions seem to be quickly accessible, which facilitates information exchange and communication.

Network analysis served as a suitable tool for making existing network structures visible in the ‘Familie+’ project in the three model regions involved. Although the networks are still in their initial stages and yet to have any influence on the project, it is important to survey and analyze the existing structures. The socio-ecological approach, which includes a multi-setting as well as participatory design and underpins the community-based obesity prevention project ‘Familie+’, is broad; therefore, the present study focused on the analysis of stakeholders from different network levels and their interaction. Interactions were captured in a manner similar to existing studies measured by type and extent of contact; however, our findings raise important questions for further research to explore community networks in a more interpretive analysis. These questions could relate to stakeholders’ barriers and contributions to communication, the rules of communication, and the dimensions and quality of interaction [28,36]. Given that obesity is a very complex problem that is difficult to map, it would be interesting to apply a systems approach that includes examining stakeholders’ views on these systems analysis methods, similar to the review by Koorts, Salmon, Swain, Cassar, Strickland, and Salmon [16]. Feedback from key stakeholders on the systems models and the resulting more accurate consideration of their needs could help to better represent how factors influence childhood obesity in different settings and at different levels within the ‘Familie^+^’ project. Nevertheless, a foundation could be created at the beginning of the project simply by pointing out previously known and unknown actors and fields of work on which the respective municipalities can build and expand their networks. A specific exchange between the stakeholders can initiate an intensive engagement with the topic of health promotion of primary school children, common goals, development opportunities, and ideas in further project phases. A project guide on the topic of the strategic design of municipal prevention and health promotion programs indicates that network analyses as part of the assessment of the current situation in the early phase of a municipal health promotion project lead to the sustainable processing of the respective topics and needs for action [38].

Considering practical implications, it is worth emphasizing the finding that the urban model region, with its very distinctive community health promotion structure, already has a well-connected network and a health facilitator. It would, therefore, be desirable to initiate the further expansion and strengthening of the network at the health policy level in the other model regions by providing the necessary resources. Given that the evidence base for network analyses in the prevention setting in particular is scarce, this study provides an initial foundation of systematic networking of the health-related network levels. Following on from this, additional health facilitators can be deployed, whose main task is to link the primary and tertiary levels. This approach has already been successfully implemented in a Brazilian family health program [14] and could also be used within ‘Familie+’ to strengthen networking and information exchange between the health policy level and primary schools, including children and their families. In addition, sufficient resources must be made available to strengthen the network so that other network and cooperation partners can be involved in a participatory manner. To achieve sustainability, it would be beneficial for practitioners to formulate checklists or handouts that function as guidelines to help other municipalities pursue interventions with the goal of obesity prevention. This process could be facilitated by a community-based approach, which has already been used in Spain to develop urban policy recommendations for obesity prevention. Using the photovoice method, all stakeholders and people affected can be involved in the research process (see Díez, et al. [39] for more information), which helps to address the complex problem of obesity more efficiently and create a healthier environment.

For further planning and a successful course of the ‘Familie^+^’ project, it would be valuable to identify potential organizational collaborations in order to maintain and strengthen partnerships and to identify any gaps in the network. Currently, there is no evidence that the use of a network analysis provides intelligent targeting of key relationships and collaborations [40]. A comparison of the networks of the three model regions over the course of the intervention would be useful to identify different multipliers that could be beneficial for knowledge transfer to the target groups.

This study also has some potential limitations in the context of preparing and conducting the network analysis. The network definition was based on the subjective results of the individual expert interviews within the respective municipalities; therefore, there may be institutions that were forgotten or otherwise excluded. The institutional contacts selected to conduct the network analysis were those that appeared to have the best knowledge of the day-to-day operations of their institutions. Still, the information provided by some respondents could be limited and inconsistent due to institutional changes (including the COVID-19 pandemic), job changes, and changes in roles and responsibilities. Another problem with conducting the survey-based network analysis was the non-participation of potentially relevant network partners in all three municipalities due to their inaccessibility by email or phone and a lack of time on the part of these stakeholders. The study is a subjective survey, implying that the results are subject to the influence of social desirability. At the municipal level, it should again be noted that the networks have different prerequisites due to their location (urban versus rural) as well as different levels of prior experience in municipal health promotion. The hierarchical position as well as the objectives of the experts and individual stakeholders are decisive in the contribution of resources that have a beneficial effect on the project ‘Familie+’ as well as on community health promotion in the model regions. Nevertheless, one strength of the study is its sufficiently high response rate, as the incomplete data sets appeared to be representative of those involved in the overall project. The inclusion of adjusted control variables as well as the identification of communication-related confounders would be advisable, as the interpretation of the results is influenced by different network sizes as well as structural characteristics or unusual events.

## 5. Conclusions

This paper outlines the application of network analysis in the context of the project ‘Familie+’, a community-based intervention for the prevention of obesity among primary school children in Germany. By mapping structural and process characteristics in three model regions, we were able to identify initial community network structures, existing relationships, and collaborations among stakeholders relevant to the project. The network models showed some similar as well as diverse characteristics, mainly due to their different sizes and the existing structures in the field of municipal health promotion. Based on the results of the network analysis, it appears that it would be useful to initialize a community health facilitator provided with the necessary resources and training at the health policy level in community health promotion projects, irrespective of the rural or urban structure of the region. Participatory engagement of relevant actors, initiated by the facilitator, may have a positive influence on the effective dissemination of information and networking with other stakeholders across the network levels. This could allow for further deciphering of the impact of network structure on community-based obesity prevention interventions.

## Figures and Tables

**Figure 1 healthcare-10-01501-f001:**
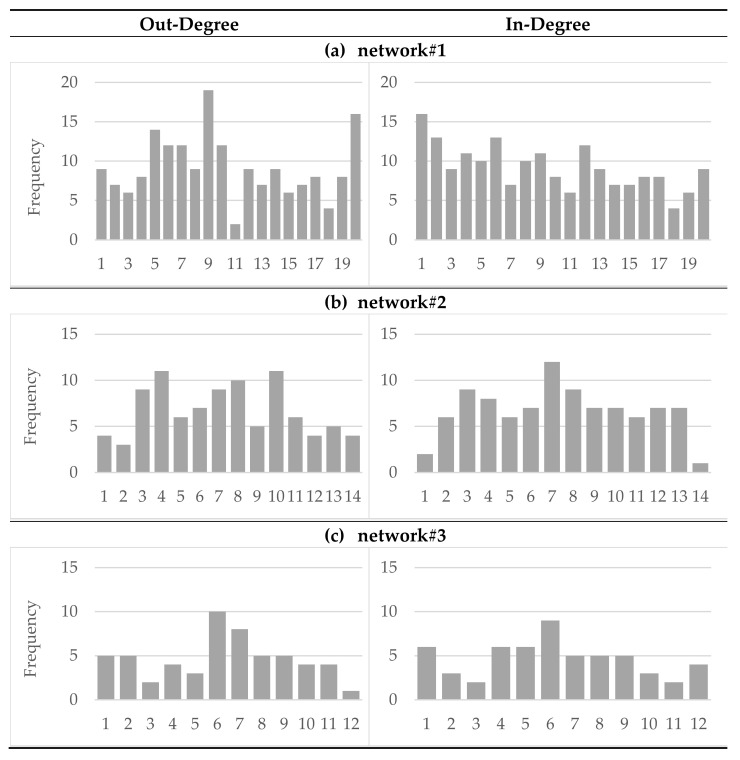
Degree distributions of all networks separately for (**a**) network#1, (**b**) network#2 and (**c**) network#3.

**Figure 2 healthcare-10-01501-f002:**
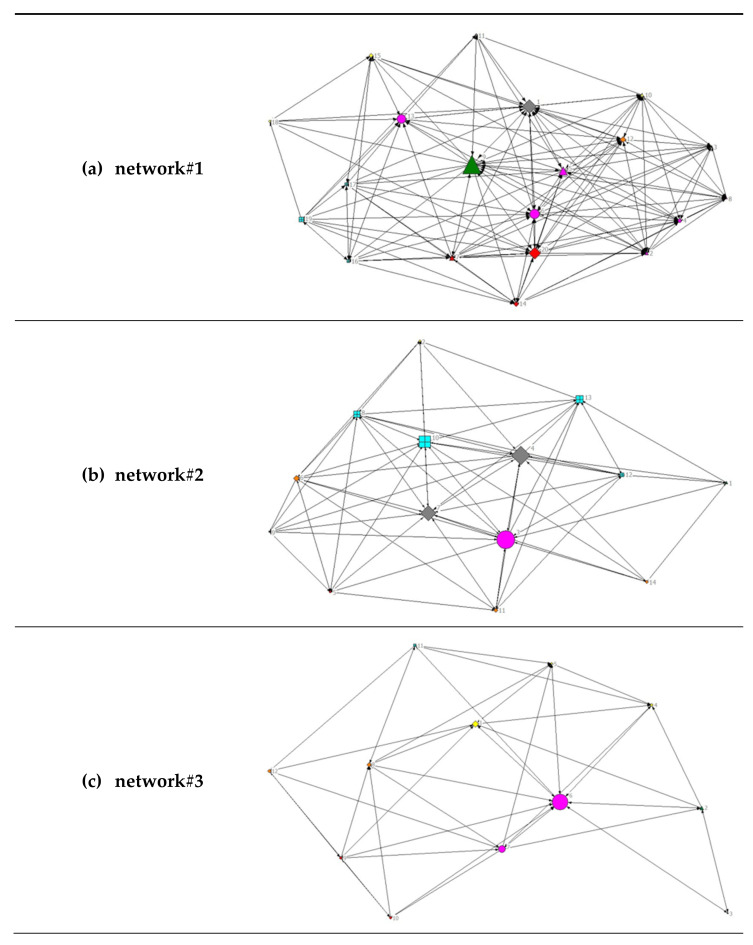
Visualization of network data for all networks. (**a**) Network#1: size = 20, density = 48%; (**b**) network#2, size = 14, density 52%; (**c**) network#3, size = 12, density 42%. Node size by betweenness. Colors by subject area: pink, health; green, nutrition; yellow, sports/exercise; orange, leisure; red, family; turquoise, elementary schools; grey, social and educational institutions. Symbols by structural level: circle, office level; up triangle, network level; square, association level; box, elementary schools.

**Figure 3 healthcare-10-01501-f003:**
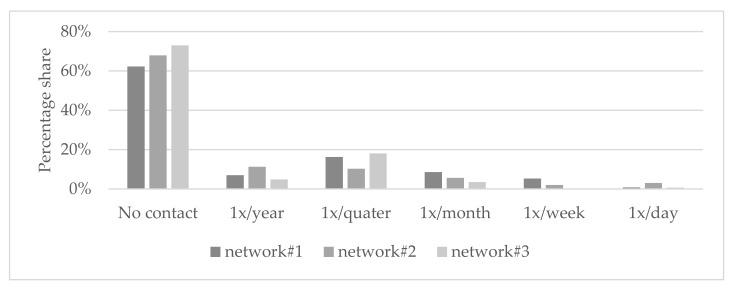
Distribution of contact frequency for all networks.

**Figure 4 healthcare-10-01501-f004:**
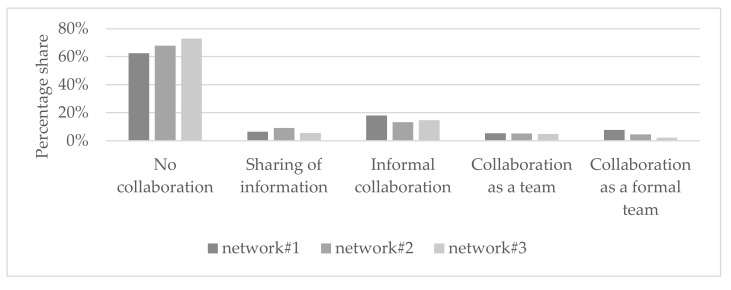
Distribution of collaboration intensity for all networks.

**Figure 5 healthcare-10-01501-f005:**
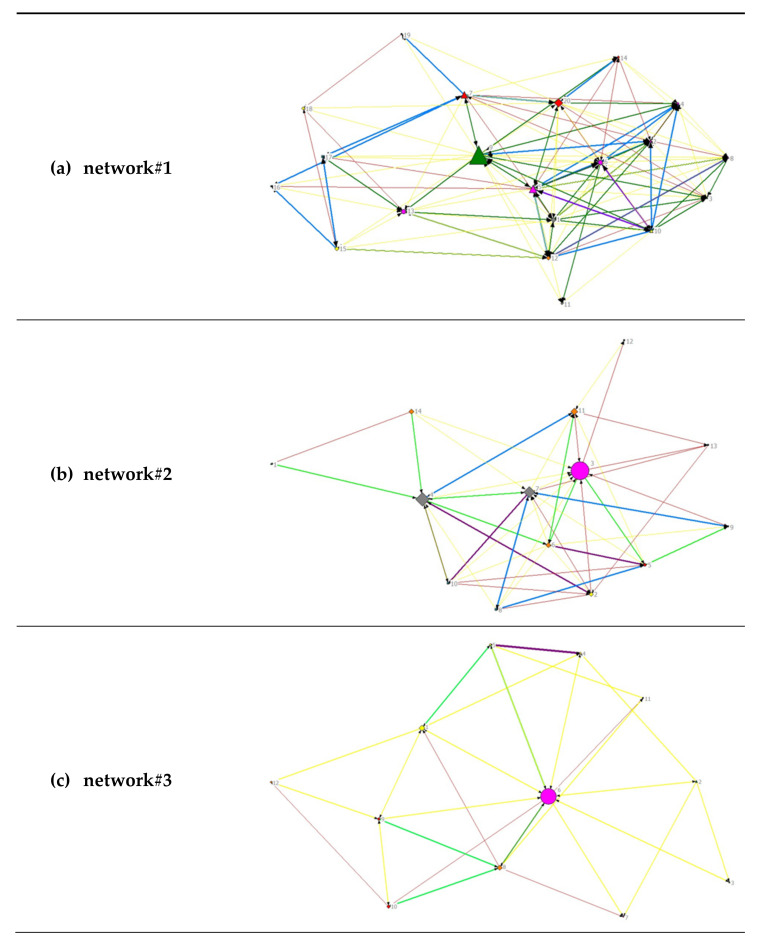
Visualization of network data regarding frequency of contacts separately for (**a**) network#1, (**b**) network#2 and (**c**) network#3. Node size by betweenness. Frequency of contact: no color, no contact; red, 1×/year; yellow, 1×/quarter; green, 1×/month; blue, 1×/week; purple, 1×/day.

**Figure 6 healthcare-10-01501-f006:**
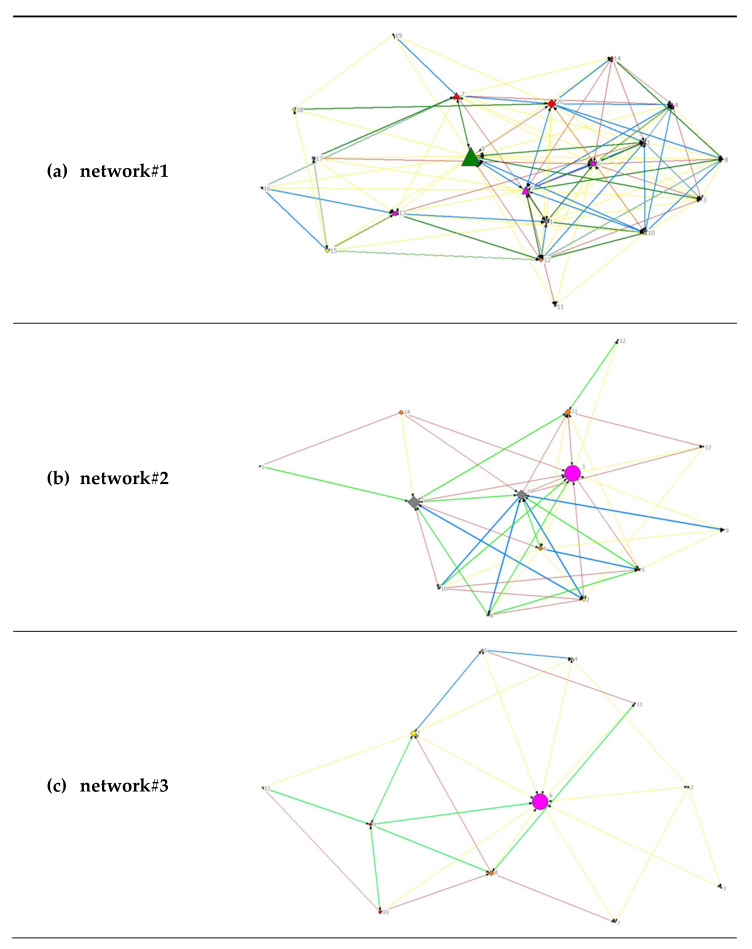
Visualization of network data regarding intensity of collaboration separately for (**a**) network#1, (**b**) network#2 and (**c**) network#3. Node size by betweenness. Intensity of collaboration: no color, no collaboration; red, sharing of information; yellow, informal collaboration to achieve common goals; green, collaboration as a team with specific responsibilities; blue, collaboration as a formal team (joint planning/sharing of staff/resources).

**Table 1 healthcare-10-01501-t001:** Subject area and structural level of the networks.

**Subject Area**	**Color**	**Stakeholders from**
**Network#1**	**Network#2**	**Network#3**
Health (superordinate level)		2, 4, 5, 6, 8, 13	3	6, 7
Nutrition		3, 9	1	2, 3
Sports/Exercise		10, 15, 18	2	1, 4, 5
Leisure		12	6, 11, 14	8, 12
Family		7, 14, 20	5	9, 10
Primary schools		16, 17, 19	8, 9, 10, 12, 13	11
Social and educational institutions (focus on health)		1, 11	4, 7	-
**Structural level**	**Symbol**	**Stakeholders from**
**Network#1**	**Network#2**	**Network#3**
Office level (higher, communally organized level)	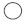	4, 6, 13	3, 5	4, 5, 6, 7, 9
Network level (networking agencies)	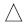	2, 3, 5, 7, 8, 9, 10	1, 2	2, 3
Association level (local level)	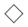	1, 11, 12, 14, 15, 18, 20	4, 6, 7, 11, 14	1, 8, 10, 12
Primary schools	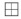	16, 17, 19	8, 9, 10, 12, 13	11

**Table 2 healthcare-10-01501-t002:** Summary of network characteristics.

	Network#1	Network#2	Network#3
	Mean	Mean	Mean
Network size	20	14	12
Number of ties	184	94	56
Contact opportunities	400	196	144
Average degree	9.20	6.74	4.67
Network density	0.48	0.52	0.42
Average distance	1.52	1.54	1.64
Degree centrality	0.57	0.39	0.58
Homophily (subject area)	0.61	0.68	0.57
Homophily (structural level)	0.41	0.32	0.25
Structural signatures defined according to Schnegg and Lang [27]: Network size: number of actors (nodes) in the network; Number of ties: number of existing connections (edges) between actors; Contact opportunities: expressed from both positions per network; Average degree: average number of links each node in the network has; Average distance: average path length among connected pairs.			

## Data Availability

The datasets used and/or analyzed during the current study are available from the corresponding author on reasonable request.

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
