# Peer review of "Network Analysis for a Community-Based School- and Family-Based Obesity Prevention Program"

_healthcare, 2022, doi:10.3390/healthcare10081501_

Round 1

Reviewer 1 Report

This research is properly structured and provides valuable information on the topic. The authors should address minor comments carefully to ensure that all issues affecting the plainness of the study are removed.

-          Abstract: It should be clear why 20, 14, and 12 stakeholders are included in the analysis of the three selected networks. The abstract should be clear and comprehensive.

-          Keywords: We suggest that the authors should replace keywords such as “Network analysis” and “obesity prevention” because these keywords are already found in the review paper title. It is better that they replace them with other keywords to increase the reach of the article.

-          Introduction Section: It contains appropriate information. However, we believe that some paragraphs should be broken down to be easier to track. Also, we recommend researchers to add more research (literary studies) on the topic and provide a clear critique of these studies.

-          Figures: Figure 2 used before Figure1 in-text.

-          English Writing: This paper requires minor proofreading of the entirety of the article to remove all the problems related to typos, spelling, and grammar mistakes. The authors should check the paper completely to verify the integrity and accuracy of the English writing

-          List of references: The references should follow Healthcare-MDPI style. Some references require updating. Some references do not contain enough information such as references [7], [12] … etc. Some search names in the references list begin with an uppercase letter in each word such as [5], [12] …etc., and other words begin with a lowercase letter, such as [1], [2] …etc. Authors should standardize the writing style of research names. This paper requires a moderate check of the reference list.

Author Response

The authors would like to thank the reviewers for their invaluable assistance and helpful comments. We noticed all annotations and revised the text accordingly. The reviewers support helps to improve the quality significantly.

Reviewer 2 Report

Dear authors,

The aim of the this study is to use network analysis as an applicable tool for evaluating stakeholder networks at the baseline of a community-based prevention intervention. With the results obtained, the network structures and possible differences of the participating model regions have mapped which may be profitable in terms of derivations for the further course of the intervention. In this paper, also network analytical methods were used to give an overview of the three model regions involved in the project Familie. Their structural and process characteristics as well as the interconnectedness of the actors within each network were outlined. The networks have mainly characterized by their different sizes as well as different distribution of network partners in terms of their subject matter and structure. To sum up, this paper outlines the application of network analysis in the context of the project Familie, a community-based intervention for the prevention of obesity among primary school children in Germany. By mapping structural and process characteristics in three model regions, initial community network structures, existing relationships and collaborations among stakeholders relevant to the project were able to be identified. The network models showed some similar as well as diverse characteristics, mainly due to their differ ent sizes and already existing structures in the field of municipal health promotion. Based on the results of the network analysis, it appears to be useful to initialize a community health facilitator provided with necessary resources and training at the health policy level in community health promotion projects, irrespective of the rural or urban structure of the region. Participatory engagement of relevant actors initiated by the facilitator can have a positive influence on the effective dissemination of information and networking with other stakeholders across the network levels. This could allow further deciphering of the impact of network structure on community-based obesity prevention interventions.

The manuscript is well-constructed in general terms and written in a very clear language. However, my discussion section felt a bit weak, so I think this section needs some improvement. I think the discussion of the manuscript will be enhanced by making a few minor addition.

Best regars!

Author Response

(The authors gave the same response as above.)
